# Preparation and Characterization of Indomethacin Supramolecular Systems with β-Cyclodextrin in Order to Estimate Photostability Improvement

**DOI:** 10.3390/molecules26247436

**Published:** 2021-12-08

**Authors:** Marzena Jamrógiewicz, Marek Józefowicz

**Affiliations:** 1Department of Physical Chemistry, Faculty of Pharmacy, Medical University of Gdańsk, Al. Gen. Hallera 107, 80-416 Gdańsk, Poland; 2Institute of Experimental Physics, Faculty of Mathematics, Physics and Informatics, University of Gdańsk, Wita Stwosza 57, 80-308 Gdańsk, Poland; marek.jozefowicz@ug.edu.pl

**Keywords:** supramolecular chemistry, cyclodextrin, inclusion complexes, photodegradation, pharmaceutical stability, indomethacin

## Abstract

Cyclodextrins have found wide application in contemporary chemistry, pharmacy and medicine. Because of their unique properties, cyclodextrins are constantly used in research on solubility or stability improvement, as well as other physicochemical properties of medicinal substances. Indomethacin (IND) is a photolabile molecule that also attracts the interest of researchers due to its therapeutic potential and the need to overcome its problematic photosensitivity. Supramolecular complexes of indomethacin with β-cyclodextrin (CD) are already known, and they show greater stability compared to complexes with other types of cyclodextrins. So far, however, the sensitivity to light of physical mixtures and inclusion complexes in the solid phase has not been studied, and their various stoichiometries have not yet been investigated. Due to this fact, the aim of the present study is to obtain supramolecular systems (inclusion complexes and physical mixtures) of indomethacin with three different amounts of β-cyclodextrin. Assessment of the photochemical stability of indomethacin-β-cyclodextrin systems in the solid state is performed in order to find the best correlation between IND stability and the amount of CD. Comparative analysis of physicochemical degradation for stoichiometry systems [CD:IND] = [1:1], [0.5:1] and [0.1:1] is performed by using ultraviolet spectroscopy, transmission—FTIR, reflection—ATR-FTIR infrared spectroscopy and DSC calorimetry.

## 1. Introduction

All medicines can cause unwanted side effects. Most side effects are due to interactions between prescription, over-the-counter and complementary medicines. Serious health problems may also occur due to the photoreactivity of active compounds [1,2]. Therefore, all possible details on drug photodegradation are always helpful in order to minimize side effects and/or optimize drug targeting by developing photoresponsive drug delivery systems. Most advanced photoprotective methods include modern technological solutions in the form of lipid systems—liposomes, niosomes and lipid nanoparticles—where the drug is dissolved in the internal water phase, while the lipid coating protects the photosensitive substance [3,4], as well as microspheres and microcapsules [5].

An attractive and frequently used method is the formation of cyclodextrin inclusion complexes with the active ingredient molecule. This allows, among other things, the protection of physicochemical labile particles, the so-called “guest”, against environmental factors to which the substance is exposed during technological processes, transport or patient use. This is one of the methods with a positive photoprotective effect on molecules [6,7,8]. For example, positive photostability studies of isradipine have been obtained by including it in a complex with β-cyclodextrin (CD). The photostability of the resulting connection was monitored using differential scanning calorimetry (DSC) and FTIR (Fourier transform infrared) spectroscopy [9]. Other research indicates that butyl methoxydibenzoylmethane, which is a UV-radiation-absorbing agent, is used as a filter in cosmetics and sunscreen preparations to improve its photostability in the complex with HP-β-CD (hydroxypropyl-β-cyclodextrin) [10]. CDs are widely used in pharmaceuticals, drug delivery systems, cosmetics and the food and chemical industries [11,12,13]. On the pharmaceutical market, there are currently many preparations containing cyclodextrins [14,15,16].

An important molecule, which is the subject of much research in the field of medical science, as well as supramolecular chemistry, is indomethacin (IND), a popular non-steroidal anti-inflammatory medicinal substance. It was officially approved by the FDA (Food and Drug Administration) in 1965. Its widespread use is mainly due to its analgesic and anti-inflammatory effects, which are used primarily in the treatment of moderate to severe pain, stiffness and swelling resulting from rheumatoid arthritis. In 1983, Pollard and Luckert demonstrated the protective effect of indomethacin in an experimental model of chemically induced colorectal cancer [17]. Indomethacin, a well-known anti-inflammatory drug and a non-selective inhibitor of cyclooxygenase-2, has been shown to exert anticancer effects in various types of cancer, including pancreatic ductal adenocarcinoma [18,19]. Many different pharmaceutical forms of this substance are registered in the world; some of these forms are powders and lyophilizates used in the preparation of solutions for intravenous injections (Table 1).

Indomethacin, despite being a known substance for many years, is still the subject of many scientific studies due to its valid therapeutic use and serious chemical instability. Studies conducted in this article concern photostability and physicochemical characteristics of indomethacin in the supramolecular systems with β-cyclodextrin. All obtained results are of great importance and provide a lot of valuable information for the improvement of the formulation of pharmaceutical products containing indomethacin as an active ingredient.

The largest amount of data on the photosensitivity of indomethacin comes from studies where this substance was analysed in the form of solutions, mainly with the use of organic solvents [20,21]. In experiments carried out on indomethacin solution, it was shown that as a result of irradiation of the solution with UV-Vis radiation, the substance degraded photochemically, resulting in the formation of eight photoproducts [22], depending on the solvent used [23,24].

The formation of supramolecular complexes with indomethacin to improve IND solubility or stability is the subject of many different publications. Complexes with indomethacin were formed to test the effect of ring size of various cyclodextrin derivatives on IND stability. α, γ and β-CD were used to show that β-CD demonstrated the strongest influence on the stability compared to other derivatives, while α-CD caused a much faster degradation of indomethacin [25]. Research confirms that β-cyclodextrin has the most appropriate cavity size for IND, which may result in better stability and even increase solubility, as was observed in previous studies of indomethacin in a liquid state. 

The novelty and the aim of this work is to investigate the photostability of indomethacin, mainly in the form of supramolecular systems as inclusion complexes (IC) and physical mixtures (PM), in the solid phase and three different stoichiometries: [1:1], [1:0.5] and [1:0.1].

## 2. Results

The guidelines of the International Council for Harmonization (ICH) of Technical Requirements for Medicines for Human Use are properly systematized and divided into thematic groups. For example, the ICH Q1B [26] standard provides information on testing the photostability of new substances and medicinal products. These guidelines indicate that the purpose of photodegradation is to assess the sensitivity of photosensitive molecules to a given stress factor, to develop an appropriate analytical method and to explain the mechanisms of degradation [27]. Research on photostability should be designed in such a way as to generate comprehensive data, enabling the validation and improvement of the relevant methodology, as well as the development of new analytical methods [28,29].

Forced degradation studies are an important part of drug development efforts, allowing a better understanding of how active substances break down [30]. This research is also helpful in the development of the most appropriate packaging for photosensitive preparations. Due to the lack of clear regulatory requirements regarding forced degradation conditions, it is suggested to use an appropriate stress factor that will ensure 10–20% degradation of a substance sensitive to a given stress factor [31,32].

Where information on potential drug substance degradation products is limited, stress testing is conducted to identify an appropriate methodology that indicates instability [29,33]. Stress tests also generate data on degradation mechanisms and possible breakdown products that may also be produced during drug substance storage. Such tests help to assess the physicochemical or stereochemical stability of medicinal substances. The degradation test procedure depends on the properties and structure of the active substance being tested and the type of final medicinal product, due to the difference in physicochemical properties of each individual compound in the preparation [34].

### 2.1. Stoichiometry of IND-β-CD Inclusion Complex in Water

The interactions between IND, β-CD and water in both ground and excited states were monitored by following the absorption and fluorescence spectral changes of the investigated molecule upon increasing β-CD concentration. As can be seen in Figure 1A,B, a gradual addition of β-CD to a solution of IND in water slightly increased the absorbance without any significant peak shift (~2 nm blue-shift). It is important to note that the steady-state area normalized absorbance spectra present an isosbestic point (see insert of Figure 1A), which indicates that two different species remain in equilibrium (IND and IND-β-CD complex in the ground state).

The interaction of β-CD with IND in the excited state was monitored by analyses of the fluorescence profile in the presence of a macrocyclic compound. As can be seen in Figure 1B, a strong increase in the fluorescence intensity with a slight blue shifting of $\lambda_{F}^{max}$ by 3−5 nm was observed when β-CD concentration was increased, keeping IND concentration constant. The relatively significant increase in fluorescence intensity upon increasing β-CD concentration may presumably be due to a decrease in nonradiative deactivation rates within the confined β-CD cavities. Moreover, presented changes in the fluorescence spectra clearly indicate a change in the microenvironment around the molecule, which might be due to the incorporation of IND inside the CD cavity.

In order to confirm the existence of the inclusion complexes in the ground and excited states—their stoichiometry and stability—a Benesi−Hildebrand plot was constructed (see Figure 1C,D). The experimental steady-state absorption and fluorescence data were used to plot the $1/\left(A-A_{0}\right)$ versus 1/[β−CD], as well as $1/\left(I-I_{0}\right)$, versus 1/[β−CD]). A linear dependence was obtained over the whole range of studied β-CD concentrations, which indicates the formation of 1:1 inclusion complexes with β-CD, both in the ground and excited states. The ground- and excited-state equilibrium constants, determined from the linear regression approach, i.e., Benesi–Hildebrand (BH) plot, were found to be $K_{1}^{g}\left(BH\right)\;$= (32 ± 2) M^−1^ and $K_{1}^{e}\left(BH\right)$ = (65 ± 4) M^−1^, respectively, and correlate well with the equilibrium constant values alternatively determined using nonlinear regression (NL) procedure $K_{1}^{g}\left(NL\right)=$ (30 ± 3) M^−1^ and $K_{1}^{e}\left(BH\right)$) = (71 ± 5) M^−1^). The $K_{1}^{g}$ and $K_{1}^{e}\;$ values determined by these two methods differ by about 7% and 9%, respectively. Moreover, these results indicate the low binding affinity of IND to β-CD in studied liquid media.

To further confirm the 1:1 stoichiometry of the formed complexes between IND and β-CD in water in the ground and excited states, a Job’s plot procedure was used. As can be seen in the insert of Figure 1C,D, the curve maximum falls at about 0.5, which confirms the 1:1 stoichiometry in the ground and excited states.

### 2.2. UV-Vis Studies of IND and Its Supramolecular Systems Photodegradation

Figure 2 displays UV–Vis absorption spectra of indomethacin after a few hours of exposure to light. The spectral shape characteristic of IND became flat after 12 h of irradiation. The absorption maximum at 318 nm was blue-shifted. This band is quite broad, and no spectacular changes are shown in Figure 2A, but Figure 2B exhibits the second derivatives of each recorded spectrum, which demonstrate significant degradation of IND.

In the study of host-guest supramolecular chemistry, absorption spectra were regularly used to verify IC formation [35,36]. Characteristics of complexes were also performed through the use of such methods as DSC [37], XRD [38,39] and NMR [40,41], with binding constants also determined [16,42]. In the present study, the absorption spectra of methanol:HCl solutions (1:9 *v*/*v*) of β-CD, IND and IC were taken into consideration for stability of medicinal compounds included in cyclodextrin. β-CD has almost no absorption throughout the wavelength scanned; consequently, its absorbance values can be neglected [43]. The absorption intensity of IND is the highest among spectrum of physical mixture and spectrum of inclusion complex (Figure 3), showing a strong hypochromic effect after mixing with β-CD. This suggests IC formation between IND and β-CD. A physical mixture of IND and β-CD prepared by mixing (shaking) in a mortar pestle also presents a similar UV spectral pattern. Inclusion-complex formation causes changes in the absorption spectra of substrates [44]. Figure 3 shows the UV–Vis spectra of the indomethacin, main stoichiometry 1:1 of inclusion complex and physical mixture. In the case of PM1, the absorption at 318 nm (indomethacin) was reduced. The changes in the UV–Vis spectra of IND in the PM and IC samples after photodegradation are presented in Figure 4 and Figure 5, respectively.

Generally, from the obtained results, it follows that either PM or IC are unstable in solutions. Additionally, powders containing indomethacin after photodegradation indicate changing UV-Vis absorption. In the case of 1:1 systems, spectra of both PM and IC (Figure 4 and Figure 5) present higher values of absorbance, but in physical mixtures, we observed a total absence of λ_max_ for IND at 318 nm. Other samples seemed to be stable just after 12 h of excitation because an even longer duration of photoirradiation does not influence the UV-Vis spectrum (PM3, IC2, IC3).

### 2.3. DSC Analysis and Specific Heat-Capacity Data

As other studies have shown [45], the curve of IND displays one sharp endothermic peak at 167 °C (440 K), which, in the case of the γ-polymorphic form, is the highest of the peaks (α and β). Table 2 presents thermodynamic data of prepared and tested samples. Endothermal melting peaks of IND in supramolecular complexes are much lower than in physical mixtures. The drug-CD physical mixture generally shows the same endothermic effect at a temperature corresponding to the melting point of the crystalline guest candidate. The interacted mixture generated by lyophilization shows a similar flat profile in a slightly lower temperature region [46]. Noncovalent bonds that arise during the formation of a complex result in positive contributions to free energy. The value of enthalpy for indomethacin in supramolecular complex 1:1 stoichiometry increases from −101.19 J/g to −0.78 and −1.31 J/g. This usually also leads to a characteristic increase in heat capacity [47]. Here, Cp IND increases from 1.08 to 1.51 J·g^−1^·°C^−1^.

Figure 6A,B show graphs of heat capacity obtained for each sample. The Cp measurements for the inclusion complexes IC1 and IC2 samples indicate that the values are a bit higher than for physical mixtures PM1 and PM2, while the value obtained for IC3 is similar to that of PM1 and PM2. It should be stressed that the results described in this paper point to the important role of the stoichiometry and the existence of proper hydrogen bonds in determining the thermodynamic balance and influence on stability of each system [48]. It might be noted that the values of Cp of IND (Figure 6C) after photodegradation also change and approve some disorder, as well as the crystal defects of indomethacin. Molecular mobility influenced by irradiation increases because higher specific heat-capacity values imply more energy obtained.

During stability testing of samples with cyclodextrins (Figure 7) the heat capacity indicates a similar distribution in the case of 1:1 and 1:0.5 systems. More significant changes intended for degradation process of IND in the system with cyclodextrin were observed in physical mixtures rather than in inclusion complexes. Melting and crystal structure of PM is probably influenced by irradiation and changed what is visible near the temperature of melting, 161–162 °C. Inclusion complexes IC1 and IC2 are stable, but the IC3 is labile and presents fluctuations of Cp values. Here, the decomposition or crystal structure destruction was observed after 24 h of irradiance. Generally, the supramolecular systems (product) were found to be chemically stable in cases with inclusions.

Additionally, what is also very interesting is that a similarity for the ΔCp graph (Figure 8) obtained for IND and physical mixture samples was observed. The highest variations of temperature range, which are responsible for molecular mobility and interactions, are indicated in cases of IC1. There are huge differences in the ΔCp in the broadest range of temperatures. At the characteristic melting point of each sample, we observed that the lowest ΔCp values are calculated (48 h and 0 h of irradiation) for IC3 (−3.72), then inversely for PM, where the highest values of ΔCp are calculated for PM3 (4.85), and the amount of CD is the lowest.

### 2.4. Verification of the Photodegradation of Indomethacin Inclusion Complexes and Physical Mixtures with β-CD by FTIR Spectrophotometry and Attenuated Total Reflection Infrared Spectroscopy (ATR-FTIR)

Photodegradation of IND in the solid state is presented as fluctuations on infrared spectra in Figure 9. Some of the bands shifted, while some protected. The vibrational characteristic of IND and its supramolecular systems are presented in Figure 10.

Due to the high noise resulting from the specificity of the reflection technique on the ZnSe crystal, the results are presented in the wavenumber range: 1800–700 cm^−1^. Generally, the band corresponding to the vibration of the -C=O bond at wavenumbers 1712 and 1687 cm^−1^, as well as 1614 and 1586 cm^−1^, is weaker just after 12 and 24 h of irradiation. Similarly, the band corresponding to the vibration of the -C=C- group at a wavenumber of 1477 cm^−1^ is weaker. The band derived from the -C-H group at a wavenumber of 1356 cm^−1^ is also flattened. Disappearance of the band at a wavenumber of 1261 cm^−1^ and the shift at 1233–1223 cm^−1^ was observed. The band derived from -C-N at wavenumbers 1187 and 1147 cm^-1^ is visible but also shifted. The -C-Cl group corresponding to the band at wavenumbers 1084 and 1065 cm^−1^ are shifted and distorted, the same as the band from the group -C-O-C- at 1025 cm^−1^.

Based on the FTIR spectra obtained (Figure 10), the presence of indomethacin in the supramolecular complexes and physical mixtures was confirmed. In each sample, the bands characteristic for indomethacin, occurring at wavelengths of 1715 cm^−1^ and 1691 cm^−1^ were assigned to the vibration of the -C=O- bond from the carboxyl group. In all systems, at a wavenumber of 1479 cm^−1^, a band was observed derived from the -C=C- bond of the aromatic IND ring.

On the spectrum of the IC1 sample, the band at 1397 is invisible. On the other hand, the band at a wavenumber of 1455 cm^−1^, which also corresponds to the stretching vibrations of the -C=C- bonds of the aromatic ring is visible in each sample. A deformation band of the -C-H bond appears at a wavenumber of 1396 cm^−1^ and only occurs on spectra of PM2, PM3 and IC3. In all physical mixtures and supramolecular complexes obtained by freeze drying with stoichiometry [1:1] and [1:0.5], there appears -C-N stretching band in the range of 1374–1355 cm^−1^. The band at wavenumbers 1261 and 1233–1223 cm^−1^ corresponding to the -C-O-C- bond of the aromatic ring occurs in each obtained system. The stretching band at a wavenumber of 1067 cm^−1^, corresponding to -C-Cl, is found only on spectrum of PM3. Numerous deformation bands -C-H appear in the range of 924–752 cm^−1^ on spectra of all systems. Either the physical mixtures or the complexes designate bands derived from β-cyclodextrin in the range of 1160–1120 (-C-O).

During the photostability tests of PM samples, an important correlation was observed between the amount of cyclodextrin and the degradation of indomethacin [32]. In samples PM2 and PM3, the disappearance of most of the bands on the FTIR spectra is visible just after 12 h and 24 h of irradiation (Figure 11). The smallest changes were observed for PM1, and only after 48 h. For example, the band corresponding to the vibration of the -C=O bond at wavenumbers 1716 and 1691 cm^-1^ disappears, as does the band corresponding to the vibration of the -C=C- bond at a wavenumber of 1455 cm^−1^. The band derived from the -C-H bond at a wavenumber of 1355 cm^−1^ also slightly changes.

The spectra of PM2 and PM3 samples show that the band corresponding to the vibration of the -C=O bond at a wavenumber of 1716 cm^−1^ does not change, while at 1691 cm^−1^, it is clearly weaker and shifted. Vibration of the -C=C- bond at a wavenumber of 1455 cm^−1^ is clearly weaker and shifted, while the band derived from the -C-H bond at a wavenumber of 1355 cm^−1^ is also distorted. The -C-N band at wavenumbers 1187 and 1147 cm^−1^ disappears, and the -C-Cl binding bands at a wavenumber of 1085 cm^−1^ are weaker and shifted.

ATR-FTIR spectra of IC samples (Figure 12) present many fewer changes after irradiation than the PM spectra. The band corresponding to the vibration of the -C=O bond at wavenumbers 1716 and 1691 cm^−1^ in samples IC2 and IC3 is clearly flattened. There are also no visible bands derived from indomethacin in the 1300–700 cm^−1^ range.

## 3. Materials and Methods

### 3.1. Materials

The tested indomethacin was purchased from Sigma-Aldrich (serial no. 065M4173V, Sigma Aldrich, Poznań, Poland). β-cyclodextrin ID: CY-2001, serial no. CYL-2518, was purchased from CYCLOLAB (Budapest, Hungary). All other chemicals were of analytical grade or higher.

### 3.2. Supramolecular Systems Preparation

The weighed indomethacin (2.29 mmol–2.6 g) was suspended in 70 mL of water and β-CD in 50 mL of water. Table 3 shows the amounts of the components. The dispersion was then placed on a magnetic stirrer with vigorous stirring at room temperature. After stirring for several minutes, the indomethacin solution was gradually added dropwise over about 15 min to the β-CD solution. The combined solutions were left on the magnetic stirrer with continued stirring for 72 h at room temperature. In turn, the obtained mixtures were filtered through a filter paper, and the filtrate was frozen in a freezer at −80 °C for 24 h. The systems were lyophilized.

In turn, physical mixtures of indomethacin PM with β-CD were prepared, using the same amounts of substances as in complex formation. After weighing out IND and β-CD, the substances were placed in a flat-bottomed flask and mixed manually by making several rotations on the axis. The system was poured through a sieve with a pore size of 800 µm. Prepared samples were assigned, as presented in Table 4.

### 3.3. Photostability Testing

In order to induce photodegradation of indomethacin, including in mixtures with cyclodextrins (PM and IC), a Suntest CPS+ Atlas chamber (Accelerated Tabletop Exposure Systems, Atlas, Gelnhausen, Germany) equipped with a xenon lamp (1.1 to 1.5 kW) and two filters—special window glass and Solar ID65—was applied. Exposure behind window glass is accurate for photostability testing (indoor indirect daylight) of pharmaceutical products according to ICH guideline “Photostability testing of new drug substances and products”. The illuminance was set at the value of exposure power 500 [W m^−2^].

Powdered samples containing IND (0.2 g) were placed in a 20 mL glass vial. In parallel with the photostability testing, a second vessel with the same sample was placed in an exposure chamber but kept in the dark (covered tightly with aluminium foil); this sample was used as a dark/blank. Both samples were irradiated. The irradiation tests were carried out at a temperature of 45 °C. Independently, the adhesive tapes coated with powders of the systems in all prepared stoichiometry were exposed to irradiation and were analysed by the ATR-FTIR method. The blank samples were exposed to radiation simultaneously and were prepared by covering the vessels with pellets and adhesive tape with aluminium foil to hide the light.

Two types of samples were evaluated the same way: free IND and supramolecular systems after irradiation for 12 h, 24 h and 48 h.

In order to test the stability of IND in the solution, 25 mg of the indomethacin substance was weighed on an analytical balance on a tared weighing paper. For irradiation, 2 mL of the stock solution at a concentration of 0.25 mg/mL was withdrawn and quantified successively by means of an automatic pipette into a quartz glass dish and a blank vial. The caps of the vessels were secured with parafilm, while the blank vial was also wrapped with aluminium foil. The samples were transferred to a Suntest CPS+ chamber and irradiated in the same conditions as the powders.

The time sequence for irradiating all sample solutions, both in glass and quartz vessels, ranged from 1 h to 6 h. Additionally, three measurements were made after irradiation for a period of 12 h.

### 3.4. Stoichiometry of Ind-β-CD Inclusion Complex in Liquid System

To determine the inclusion complex stoichiometry and equilibrium constants in the ground (g) and excited (e) states, the changes in the steady-state absorption and fluorescence data were analysed by the Benesi-Hildebrand equation [49]:(1)$$\frac{1}{I-I_{0}}=\frac{1}{K_{n}^{e\;}\left(I_{1}-I_{0}\right)}\cdot \frac{1}{{\left[CD\right]}^{n}}+\frac{1}{I_{1}-I_{0}}\;\text{and}\;\frac{1}{A-A_{0}}=\frac{1}{K_{n}^{g}\left(A_{1}-A_{0}\right)}\cdot \frac{1}{{\left[CD\right]}^{n}}+\frac{1}{A_{1}-A_{0}}$$
where *n* is inclusion complex stoichiometries, *I*_0_ (*A*_0_), *I* (*A*) and *I*_1_ (*A*_1_), are the fluorescence intensities (or absorbances) in the absence of, at intermediate concentration and at infinite concentration of *CD*. Thus, 1/(*I* − *I*_0_) or 1/(*A* − *A*_0_) should follow a linear dependence on 1/[*CD*]*^n^* for the correct stoichiometry (*n*).

Several experimental studies [50,51,52] have shown that nonlinear least-squares regression analysis can be considered much more accurate for determination of the equilibrium constant value than the Benesi-Hildebrand procedure. When only 1:1 complex is formed, the ground- and excited-state equilibrium constants can be calculated by using the following relations:(2)$$I=\frac{\left(I_{0}+I_{1}K_{1}^{e}{\left[CD\right]}_{0}\right)}{\left(1+K_{1}^{e}{\left[CD\right]}_{0}\right)}\;\text{and}\;A=\frac{\left(A_{0}+A_{1}K_{1}^{g}{\left[CD\right]}_{0}\right)}{\left(1+K_{1}^{g}{\left[CD\right]}_{0}\right)}$$

The inclusion complex stoichiometry can be also determined from the most popular method used in supramolecular host-guest chemistry—Job’s continuous variation method [53]. Job plots were generated by plotting ΔA·R (or ΔI·R) against R, where ΔA (ΔI) is the difference in absorbance (fluorescence intensity) of the investigated organic molecule, M (Ind), without and with β-CD and R = [M]/([M] + [β-CD]).

Steady-state absorption and fluorescence measurements in water were carried out using a computer-controlled Shimadzu UV-2401 PC spectrophotometer (Shimadzu, Kyoto, Japan) and a Shimadzu RF-5301 PC spectrofluorometer (Shimadzu, Kyoto, Japan). Fluorescence was observed perpendicular to the direction of the exciting beam. Thus, errors due to fluorescence reabsorption were reduced in such a way that mathematical corrections were superfluous. The luminescence spectra were corrected for the spectral response of the photomultiplier Hamamatsu R-928 (1126 Ichino-Cho, Hamamatsu City, Japan).

### 3.5. UV-Vis Studies

UV–Vis spectra of the samples were recorded against blanks in the 200–350 nm wavelength range according to European Pharmacopoeia [54] on a Shimadzu UV-1800 spectrophotometer (Shimadzu, Kyoto, Japan). The recommended concentration for measurements using the UV-Vis method for indomethacin is 0.025 mg/mL in a mixture of 1 volume of hydrochloric acid (1 mol/L) and 9 volumes of methanol. To achieve the results presented in Figure 3, a concentration 0.01 mg/mL of IND solutions was applied.

### 3.6. ATR-FTIR Spectroscopy Application for Verification of the Photodegradation of Indomethacin Inclusion Complexes and Physical Mixtures with β-CD

For ATR-FTIR analysis of inclusion complexes, physical mixtures and free IND thin adhesive tapes were applied and covered with a layer of each powder separately. The tape strips were irradiated in a Suntest CPS+ chamber. The subsequent degradation process was analysed on a Jasco FTIR-4700 spectrophotometer (JASCO, Tokyo, Japan) using the ATR ZnSe attachment. Before each measurement, the ATR crystal was cleaned with 2-propanol. The sample placed on the crystal was mounted on the ATR adapter, and the bridge of the adapter was lowered and blocked so that the tape with the sample was pressed against the crystal. The spectra of all the samples were recorded in this way.

### 3.7. The Thermal Behaviour of the IND Supramolecular Systems

The thermal behaviour of the IND samples was measured by differential scanning calorimetry (DSC), STAR-1 System (Mettler Toledo, Greifensee, Switzerland), which was calibrated with indium at a heating rate of 10 K min^−1^. The thermal behaviour was studied by heating 2–10 mg samples in aluminium crimped pans. Heat capacities of supramolecular and free IND samples were also measured by Mettler Toledo Star One Differential Scanning Calorimeter (DSC) (Mettler Toledo, Greifensee, Switzerland). During the measurement, an inert atmosphere was created under a nitrogen flow of 60 mL min^−1^. The sapphire method for Cp determination was used [55]. A “baseline” or blank measurement was performed at a heating rate of 10 K min^−1^. All results obtained were blank-curve corrected and performed twice. The test material and the reference were placed into individual aluminium crucibles, which were then sealed with pierced lids. The data from the DSC were recorded and then analysed to obtain the Cp. ΔCp values were calculated by subtracting Cp (48 h) and Cp (“0” h) in the whole temperature range.

## 4. Conclusions

Pharmacy is inextricably linked, i.e., thematically and in terms of workshops, with the field of chemistry, including supramolecular chemistry. The greatest merit in the protection of molecules of medicinal substances against degradation has already been achieved. In this study, indomethacin systems with β-CD were tested in order to monitor the interaction and effect of β-cyclodextrin on the photostability of the drug substance. Various stoichiometries of both physical and supramolecular connections, as IND:CD [1:0.5], [1:0.1] and [1:1], were obtained and identified in order to study the effects of the amount of cyclodextrin in the system on the stability of indomethacin. Effects of the indomethacin photoexcitation process were hard to clearly identify through analytical methods, especially in the solid state, but also in a solution. In spite of this, this configuration of IND samples had not been evaluated yet. The research shows that photodegradation in systems with different stoichiometry occurs in different ways, but cyclodextrins improve the molecular structure and stability of indomethacin. Vibrational spectroscopy, especially ATR-FTIR, approve the photostability of IND because the spectra of inclusion complexes are less changing during the time of irradiation. Each of the spectra recorded for pure IND and PM after photodegradation presents fluctuations and shifts, which approve the chemical degradation. In the case of IC, there were fewer changes observed, which may suggest improvement of IND stability in the supramolecular complex with cyclodextrin.

Besides, determined heat capacity values of physical mixtures present more fluctuations, similarly to pure IND, while obtained values (and graphs) for inclusion complexes suggest that is quite a stabile system during photodegradation. The greater the amount of CD, the fewer the fluctuations that are observed.

In spite of the instability of IND supramolecular systems in solutions, we claim that the results are valuable to further research regarding the influence of different solvents on the stability of complexes. 

## Figures and Tables

**Figure 1 molecules-26-07436-f001:**
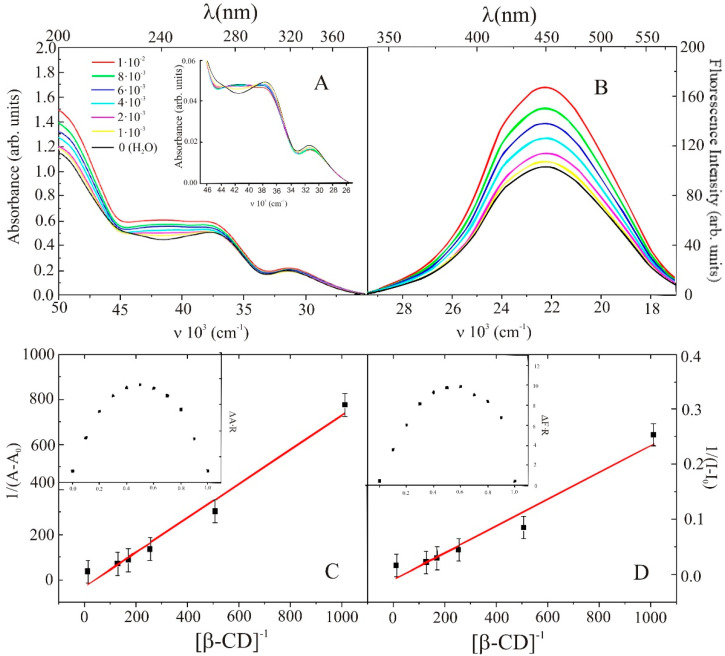
Absorption (**A**) and emission (**B**) spectra of indomethacin (5×10^−5^ M) in water containing different concentrations of β-CD, from 0 M to 10^−2^ M. Ground- (**C**) and excited-state (**D**) Benesi–Hildebrand dependence. Insert: (**A**) normalized absorption spectra of IND in water containing different concentrations of β-CD; job plots of the IND-β-CD system prepared by using steady-state absorption (**C**) and emission (**D**) data.

**Figure 2 molecules-26-07436-f002:**
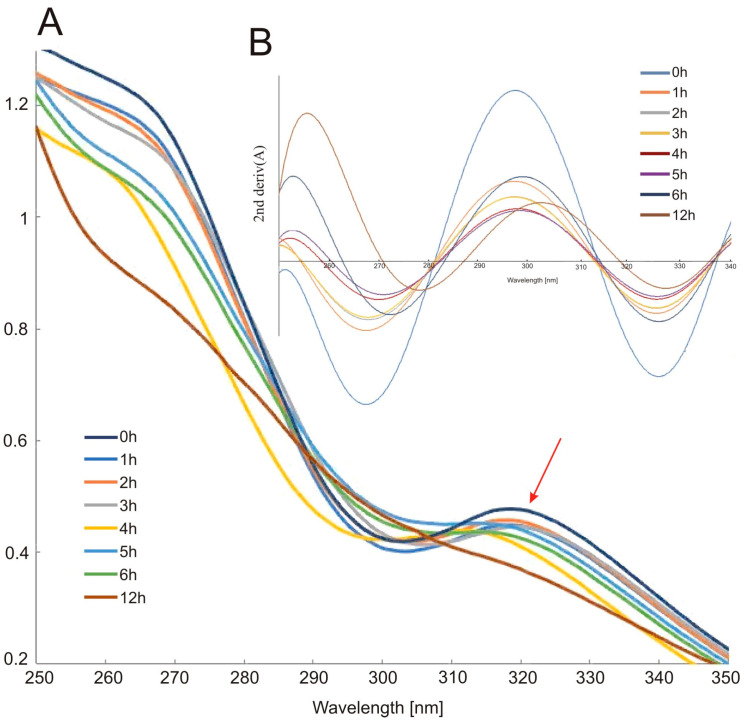
UV-Vis spectra of IND solution samples after photodegradation of (**A**) raw spectra and (**B**) second-derivative of spectra.

**Figure 3 molecules-26-07436-f003:**
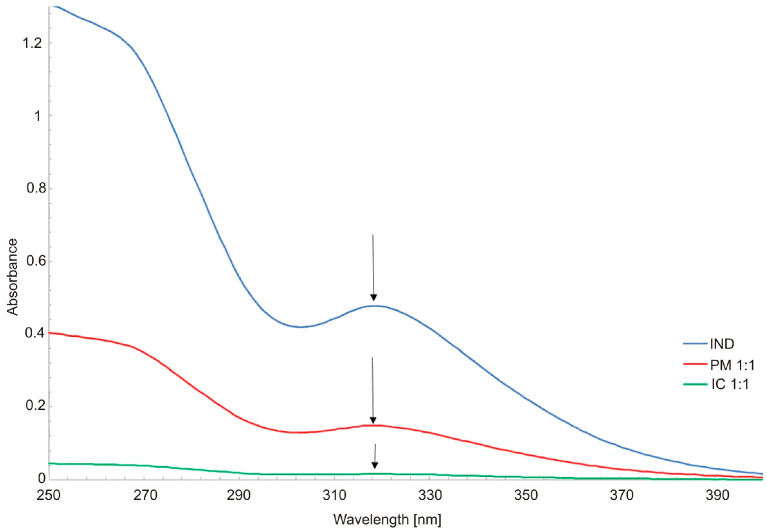
UV-Vis spectra of free IND (blue) and its supramolecular systems in stoichiometry 1:1, physical mixture (red) and inclusion complex (green).

**Figure 4 molecules-26-07436-f004:**
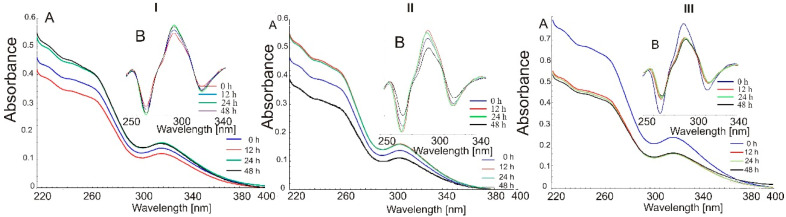
(A) UV-Vis spectra; (B) second derivatives of UV-Vis spectra of physical mixtures: (**I**) PM1, (**II**) PM2 and (**III**) PM3 after photodegradation: 0 h (blue), 12 h (red), 24 h (green) and 48 h (black).

**Figure 5 molecules-26-07436-f005:**
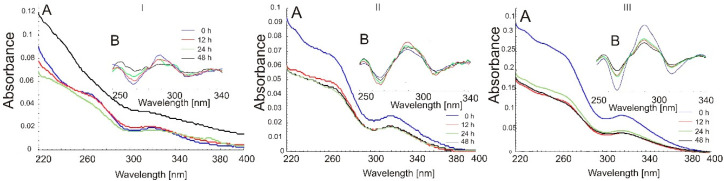
UV-Vis spectra of inclusion complexes (blue), (**I**) IC1, (**II**) IC2 and (**III**) IC3 (blue) after photodegradation: 12 h (red), 24 h (green) and 48 h (black).

**Figure 6 molecules-26-07436-f006:**
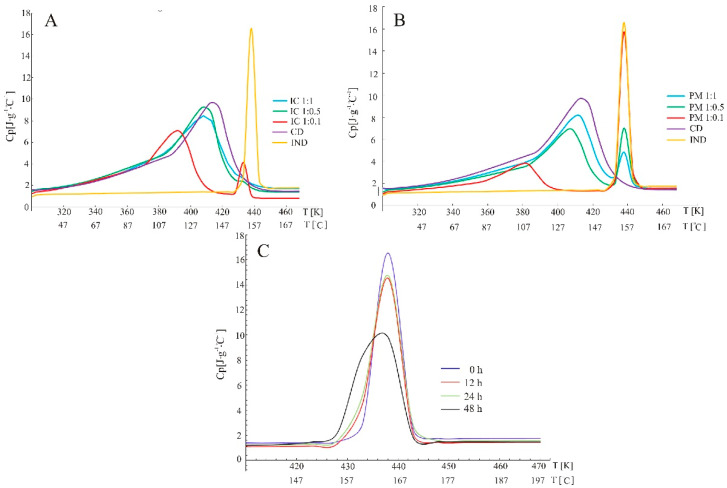
Heat capacity of IND, cyclodextrin, inclusion complexes (**A**) and physical mixtures (**B**), as well as free IND after photoirradiation as a function of temperature (**C**).

**Figure 7 molecules-26-07436-f007:**
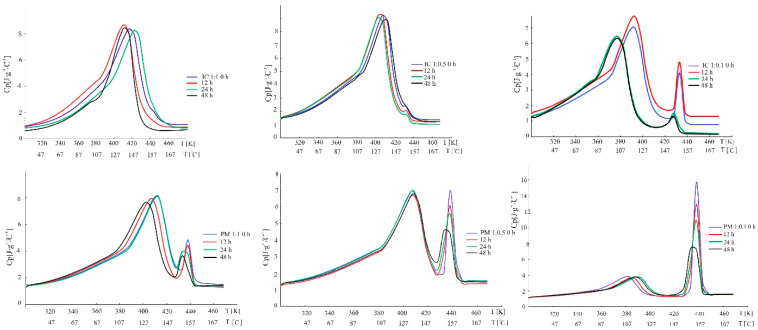
Heat capacity of physical mixtures (upper graphs) and inclusion complexes (lower graphs) (blue) in order, 1:1, 1:0.5 and 1:0.1, respectively, after photoirradiation for 12 h (red), 24 h (green) and 48 h (black) as a function of temperature.

**Figure 8 molecules-26-07436-f008:**
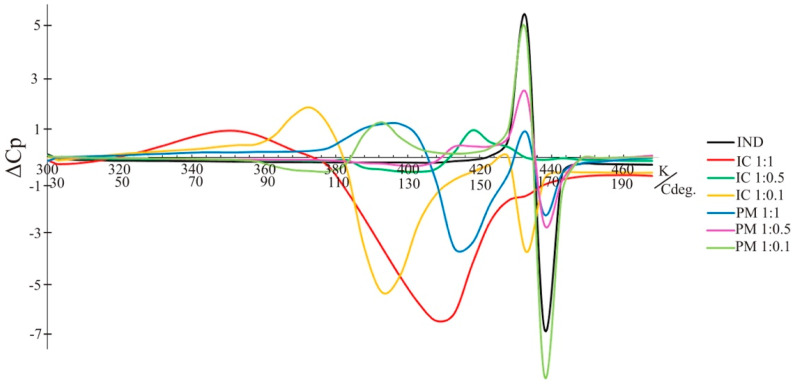
Variations of heat capacity differences after photoirradiation, depending on the temperature, obtained for IND and its samples in a form of supramolecular systems.

**Figure 9 molecules-26-07436-f009:**
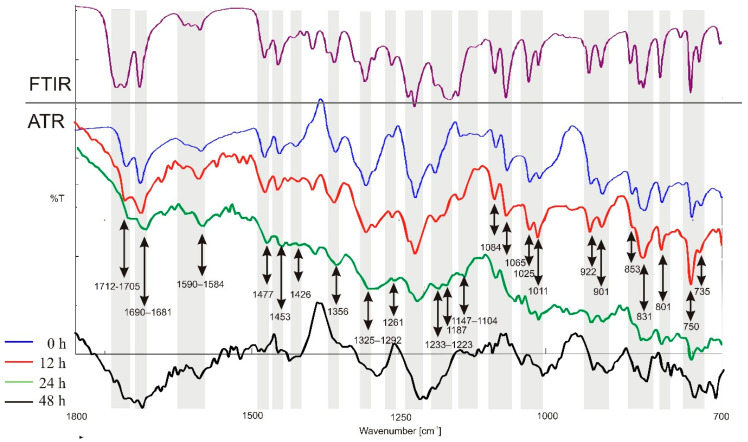
Vibrational, infrared spectra recorded for IND by transmission (FTIR—upper, violet) and reflection (ATR—lower, blue) techniques for samples after photoirradiation for 12 h (red), 24 h (green) and 48 h (black).

**Figure 10 molecules-26-07436-f010:**
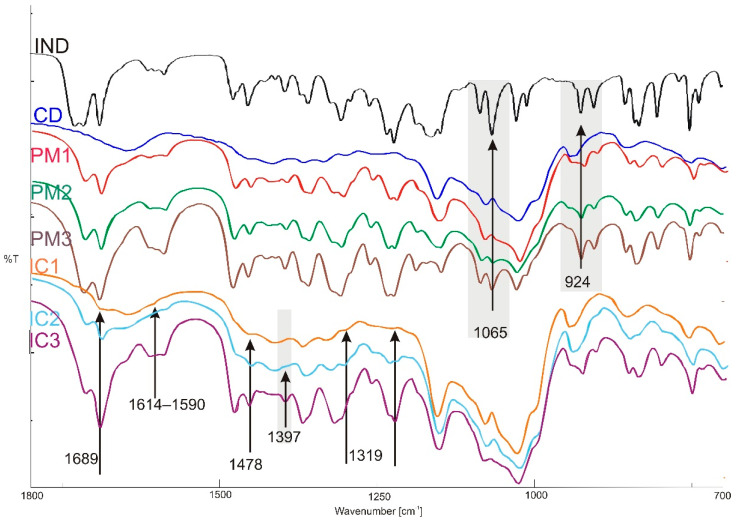
Vibrational FTIR spectra recorded for supramolecular complexes and physical mixtures of IND and β-cyclodextrin.

**Figure 11 molecules-26-07436-f011:**
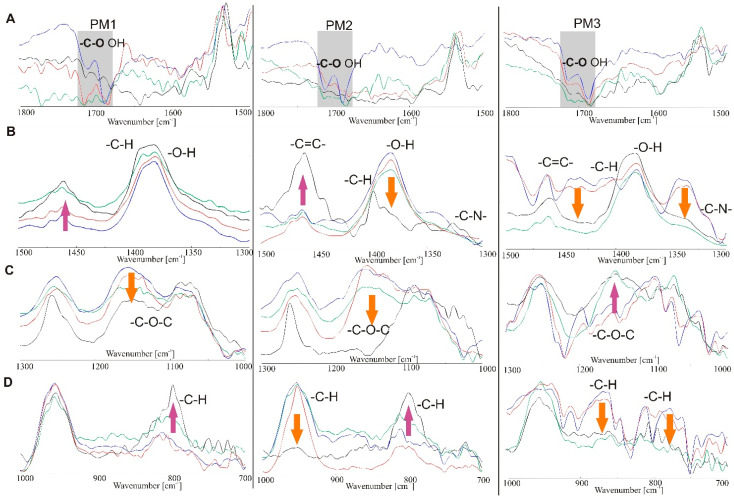
Vibrational ATR-FTIR spectra recorded for physical mixtures of IND and β-cyclodextrin (blue) after 12 h (red), 24 h (green) and 48 h (black) at ranges of (**A**) 1800–1500 cm^−1^, (**B**) 1500–1300 cm^−1^, (**C**) 1300–1000 cm^−1^ and (**D**) 1000–700 cm^−1^.

**Figure 12 molecules-26-07436-f012:**
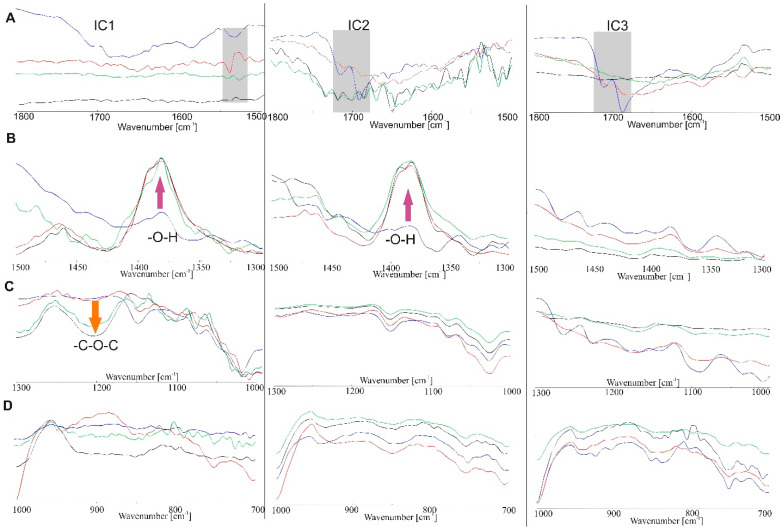
Vibrational ATR-FTIR spectra recorded for inclusion complexes of IND and β-cyclodextrin (blue) after 12 h (red), 24 h (green) and 48 h (black) at ranges of (**A**) 1800–1500 cm^−1^, (**B**) 1500–1300 cm^−1^, (**C**) 1300–1000 cm^−1^ and (**D**) 1000–700 cm^−1^.

**Table 1 molecules-26-07436-t001:** Medicines containing indomethacin as an active substance available at pharmacies.

Preparations	Pharmaceutical Form of the Preparation
Chrono-Indocin (Merck Sharp & Dohme-Chibret, France)	Capsules 75 mg
Indocid, (MSD, China)	Capsules 75 mg
Indocin (Iroko Pharms, USA)	Oral suspension 25 mg/5 mL–bottle 237 mL
Indocin (Lundbeck, USA)	Powder for injection solution (sodium salt)–vial 1 mg
Indomethacin (Avanthi, USA)	Prolonged release capsules 75 mg
Indomethacin (Fresenius Kabi, USA)	Lyophilized substance solution for injection–vial 1 mg
Indomethacin (G & W Labs, USA)	Suppositories 50 mg
Indomethacin (Ivax; Mylan; Sandoz, USA)	Capsules 25 mg and 50 mg
Indomet-ratiopharm (Ratiopharm, Denmark)	Hard capsules 25 mg and 50 mg Capsules retard 75 mg Suppositories 50 mg and 100 mg Gel 1%—50 mg and 100 mg
Metindol (Glaxosmithkline Pharmaceuticals, Poland)	Ointment 0.05 g/g|30 g
Elmetacin (Stada Arzneimittel AG, Germany)	Aerosol 0.01 g/g|50 mL

**Table 2 molecules-26-07436-t002:** Thermodynamic data of prepared and tested samples.

Sample	Molar Ratio IND:CD	Tm [°C, K]	Enthalpy [J/g]	Cp at 298 K [J⋅g^−1^⋅°C^−1^]	ΔCp at T_m_
IND	-	167.5; 440.5	−101.19	1.08	−6.74
CD	-	290–300; 563–573	not tested	1.44	-
IC1	1:1	151.98; 424.98 160.99; 433.99	−0.78; −1.31	1.51	−2.52; −1.51
IC2	1:0.5	151.62; 424.62	−3.33	1.52	0.5
IC3	1:0.1	160.75; 433.75	−19.57	1.32	−3.72
PM1	1:1	161.76; 434.76	−26.72	1.35	0.99
PM2	1:0.5	161.13; 434.13	−32.58	1.33	2.52
PM3	1:0.1	162.65; 435.65	−80.19	1.21	4.85

**Table 3 molecules-26-07436-t003:** Amounts of components used for preparation of supramolecular systems.

	Stoichiometry CD:API	β-CD	IND	Mass of Obtained Powder
I.	1:1	2.29 mmol (2.6 g)	2.29 mmol (0.819 g)	2.04 g
II.	0.5:1	1.145 mmol (1.3 g)	2.29 mmol (0.819 g)	1.30 g
III.	0.1:1	0.229 mmol (0.26 g)	2.29 mmol (0.819 g)	0.40 g

**Table 4 molecules-26-07436-t004:** Assignation of samples obtained and intended for stability testing.

Components	[IND:CD] = [1:1]	[IND:CD] = [1:0.5]	[IND:CD] = [1:0.1]
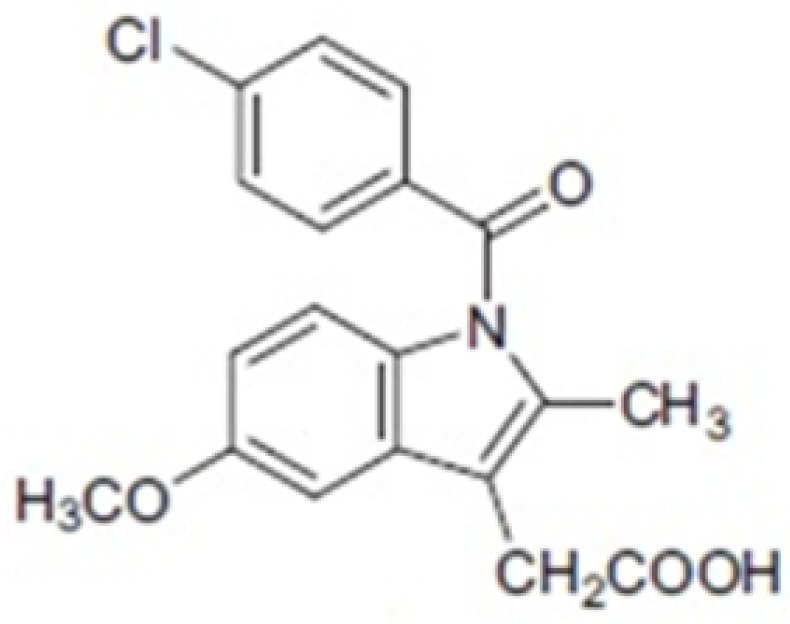	IC1	IC2	IC3
+			
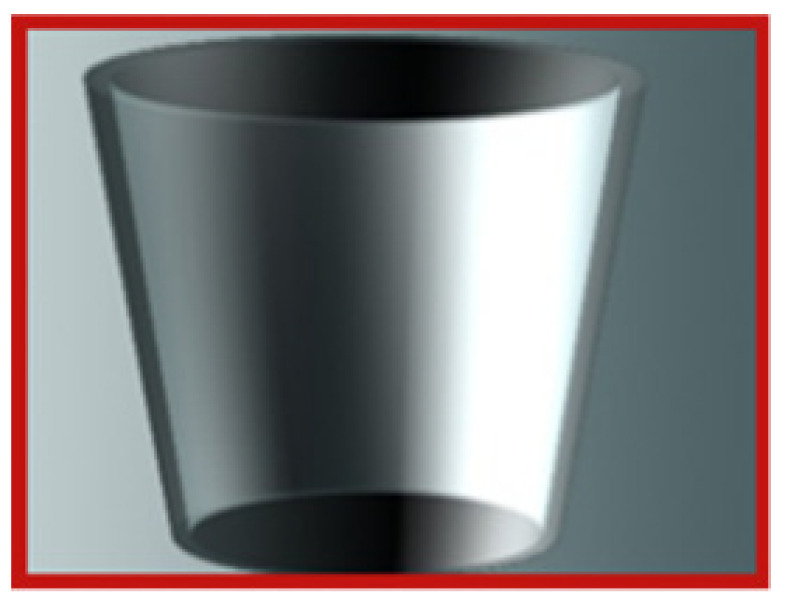	PM1	PM2	PM3

## Data Availability

The data presented in this study are available on request from the corresponding author. The data are not publicly available due to the lack of requirements of Medical University of Gdansk.

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
