# Peer review of "Preparation and Characterization of Indomethacin Supramolecular Systems with β-Cyclodextrin in Order to Estimate Photostability Improvement"

_molecules, 2021, doi:10.3390/molecules26247436_

Round 1

Reviewer 1 Report

The manuscript molecules-1480720 "Preparation and characterization of indomethacin supramolecular systems with β-cyclodextrin in order to estimate photostability improvement" by Jamrógiewicz and Józefowicz describes the study of photostability of supramolecular complexes of indomethacin with β-cyclodextrin. The paper will definitely be of interest to the readers of Molecules.

Questions and comments:

1) Since the complexes of indomethacin with cyclodextrins have been studied for a long time (10.1016/j.xphs.2021.07.002, 10.1016/S0031-6865(97)00003-4 etc.), how do the characteristics of the complex (constant, etc.) relate to the literature?

2) Why was the study of complexation investigated in water (caption of Figure 1), and UV experiments in a mixture of methanol-HCl? Moreover, the experimental part contains information about the water: ethanol = 1: 1 [v / v] system.

3) Figure 3. Please add experimental conditions (concentration of compounds, solvent etc.).

4) In modern supramolecular chemistry, methods for calculating the binding constants using graphs (like Absorbance vs Concentration) are rarely used due to their large error. I ask the authors to evaluate the binding constants using program for calculations of Host-Guest interaction BindFit (http://app.supramolecular.org/bindfit/).

5) The text in Figures 4, 5, 7, 11, 12 is hard to see. I suggest that the authors enlarge these Figures.

6) In the captions of Figures 4-6, add the decoding of letters A, B, C.

7) There is a problem with superscripts and subscripts in the manuscript, for example cm-1.

8) The authors cite a small number of articles over the past 5 years. I recommend that the authors strengthen the Introduction part on the design of cyclodextrines based supramolecular systems. Recent research/review articles on cyclodextrin applications should be added.

Author Response

Dear Professor,

I would like to express my thanks for Your assistance and opportunity to revise some parts of the manuscript entitled “Preparation and characterization of indomethacin supramolecular systems with β-cyclodextrin in order to estimate photostability improvement” by Marzena Jamrógiewicz and  Marek Józefowicz.

Please, refer to our modifications prepared in order to the comments. Language has also been approved. Performed corrections are assigned in yellow.

I sincerely thank for the time and effort in reviewing and correcting the manuscript and providing such important remarks, which definitely help to clarify the importance of this paper.

I hope that the revision will meet Yours and Editor’s expectations.

Yours sincerely,

Marzena Jamrógiewicz, PhD, D.Sc.

Explanations

  1. Since the complexes of indomethacin with cyclodextrins have been studied for a long time (10.1016/j.xphs.2021.07.002, 10.1016/S0031-6865(97)00003-4 etc.), how do the characteristics of the complex (constant, etc.) relate to the literature?

Thank to that question we add the information into the text, in line 175 “Characteristics of complexes were also performed by the usage of different methods, such as DSC [10.1016/j.xphs.2021.07.002], XRD [10.1016/S0031-6865(97)00003-4, 10.1016/j.ijpharm.2015.01.010], NMR [Pessine, F. B., Calderini, A., & Alexandrino, G. L. (2012). Cyclodextrin inclusion complexes probed by NMR techniques. Magnetic resonance spectroscopy, 3, 264, Jamrógiewicz, M., Wielgomas, B., & Strankowski, M. (2014). Evaluation of the photoprotective effect of β-cyclodextrin on the emission of volatile degradation products of ranitidine. Journal of pharmaceutical and biomedical analysis, 98, 113-119.] and binding constants are also determined [10.1016/j.ijpharm.2003.10.012, 10.1016/j.ijpharm.2006.10.044]”.

  1. Why was the study of complexation investigated in water (caption of Figure 1), and UV experiments in a mixture of methanol-HCl? Moreover, the experimental part contains information about the water: ethanol = 1: 1 [v / v] system.

There are special recommendations in pharmacy, so we applied procedure described in European Pharmacopoeia [European Pharmacopoeia 9.0, Indomethacin, 01/2017:0092, 2017].

Identification method for Indomethacin is UV-Vis spectrophotometry and proposed solution of methanol-HCl. We apologize for the editorial mistake about the water: ethanol = 1: 1 [v / v] system. We used this method and solvents in another work. Here, currently we prepared complexes in water.

  1. Figure 3. Please add experimental conditions (concentration of compounds, solvent etc.).

Thank you for the remark. Final concentration of IND solutions was 0.01 mg / ml. We add the information in line 405.

  1. In modern supramolecular chemistry, methods for calculating the binding constants using graphs (like Absorbance vs Concentration) are rarely used due to their large error. I ask the authors to evaluate the binding constants using program for calculations of Host-Guest interaction BindFit (http://app.supramolecular.org/bindfit/)

The following paragraph providing more details on determination of equilibrium constants has been added to the 3.4 section. Several experimental studies [51-53] have shown that nonlinear least-squares regression analysis can be considered much more accurate to determine equilibrium constant value than the Benesi-Hildebrand procedure. When only 1:1 complex is formed, the ground and excited state equilibrium constants can be calculated by using the following relations:     (2a)      (2b)

The new sentences providing more details has been added to the 2.1 section.

The ground and excited state equilibrium constant, determined  from the linear regression approach i.e., Benesi–Hildebrand (BH) plot, was found to be 32±2) M−1 and 65±4) M−1, respectively,  and correlates well with the equilibrium constant values alternatively determined using nonlinear regression (NL) procedure ( 30±3) M−1 and (71 ± 5) M−1). The  and values determined by these two methods differ by about 7% and 9%, respectively. Moreover, these results indicate the low binding affinity of IND to β-CD in studied liquid media.

The error bars have been added to Fig. 1C and Fig. 1D

  1. The text in Figures 4, 5, 7, 11, 12 is hard to see. I suggest that the authors enlarge these Figures

Thank you for this remark. Now, the figures are improved.

  1. In the captions of Figures 4-6, add the decoding of letters A, B, C.
  2. Thank you for this remark. The captions are improved now.
  3. There is a problem with superscripts and subscripts in the manuscript, for example cm-1.

Thank you, we’ve improved them.

  1. The authors cite a small number of articles over the past 5 years. I recommend that the authors strengthen the Introduction part on the design of cyclodextrines based supramolecular systems. Recent research/review articles on cyclodextrin applications should be added.

We agree with the statement and we added the sentence as well as above 20 new citations from the last 5 years. “CDs are widely used in pharmaceuticals, drug delivery systems, cosmetics, and the food and chemical industries” in line 55 and above.

Reviewer 2 Report

This manuscript reports the studies concerning the supramolecular systems (inclusion complexes and physical mixtures) of indomethacin with different amounts of b-cyclodextrin.

This work seems to have been carefully conducted, but the results are not clearly evidenced. Which are the photostability improvements achieved with the systems using b-CD and the more relevant conclusions drawn by the Authors about their study?

Thus, in my opinion, these points should be better explained before the manuscript can be accepted for publication in Molecules. In addition, some small corrections should also be addressed:

  • Page 2, line 53: It should be “UV radiation absorbing agent, is used as” remove “and”
  • Introduction can be reduced; it is a little repetitive.
  • Page 5, line 177: after IC should be added (inclusion complex) the first time it appears in the text; the same for PM.
  • Pages 6 and 11: the quality of Figures 5 and 11 should be improved.
  • Page 7, Table 2: What is the meaning of :[Jg^-1 °C^-1]?
  • Page 8, Figure 6 caption: the letter (C) is missing.
  • Page 10, line 269 (and also other pages/lines): it should be “1800-700 cm–1” (superscript).
  • Page 13, lines 366 and 367: it should be “mL”.

Author Response

Dear Professor,

I would like to express my thanks for Your assistance and opportunity to revise some parts of the manuscript entitled “Preparation and characterization of indomethacin supramolecular systems with β-cyclodextrin in order to estimate photostability improvement” by Marzena Jamrógiewicz and  Marek Józefowicz.

Please, refer to our modifications prepared in order to the comments. Language has also been approved. Performed corrections are assigned in yellow.

I sincerely thank for the time and effort in reviewing and correcting the manuscript and providing such important remarks, which definitely help to clarify the importance of this paper.

I hope that the revision will meet Yours and Editor’s expectations.

Yours sincerely,

Marzena Jamrógiewicz, PhD, D.Sc.

Explanations

  1. This work seems to have been carefully conducted, but the results are not clearly evidenced. Which are the photostability improvements achieved with the systems using β-CD and the more relevant conclusions drawn by the Authors about their study?

Thank you for this question.

We add some explanation into the coclusion part of manuscript.

Vibrational spectroscopy, especially ATR-FTIR approve the photostability of IND because the spectra of inclusion complexes are less changing during time of irradiation. Each spectra recorded for pure IND and PM after photodegradation present fluctuations and shifts which approve the chemical degradation. In case of IC there are less changes observed what may suggest improvement of IND stability in the supramolecular complex with cyclodextrin.

Besides, determinated heat capacity of physical mixtures present more fluctuations, similarly to pure IND while obtained values (and graphs) for inclusion complexes suggest that is quite stabile system during photodegradation. The more amount of CD the less fluctuations are observed.

In spite of the instability of IND supramolecular systems in solution we clam, that the results are valuable to further research. At this stage of studying different complexes of IND with β-CD we observed that the smallest changes on UV-Vis spectra are referred to the inclusion complexes. We fell that our research in solution should be broaden and different solvents influence on the complexes stability should be verified. We used recommended by Pharmacopoeia solvents mixture for indomethacin but probably this is not the best solution for complexes with cyclodextrins.

  1. Page 2, line 53: It should be “UV radiation absorbing agent, is used as” remove “and”

We agree, thank you. It’s corrected.

  1. Introduction can be reduced; it is a little repetitive.

We tried do our best in shortage the text and we deleted lines 40-43, 62-65, 87-94.

  1. Page 5, line 177: after IC should be added (inclusion complex) the first time it appears in the text; the same for PM.

We add the information in line 104, a bit earlier. Thank you.

  1. Pages 6 and 11: the quality of Figures 5 and 11 should be improved.

We enlarged the fonts and hope you find it well.

  1. Page 7, Table 2: What is the meaning of :[Jg^-1 °C^-1]?

Thank you we exchanged the meaning of heat capacity unit into J×g-1×°C-1

  1. Page 8, Figure 6 caption: the letter (C) is missing.

Now it’s corrected, thank you

  1. Page 10, line 269 (and also other pages/lines): it should be “1800-700 cm ” (superscript).

Now it’s corrected, thank you

  1. Page 13, lines 366 and 367: it should be “mL”

Now it’s corrected, thank you

Reviewer 3 Report

The authors describe the superior photostability of IND in an inclusion complex with b-CD. From a pharmaceutical applications standpoint, the common derivatives of b-CD used are 2HP-b-CD and sulfobutylether-b-CD (Captisol®). Although the present work doesn’t include the above-mentioned derivatives of b-CD, the photo stabilization studies carried out might bear impact for the work to be accepted for publication after addressing the following points: -

1) The novelty of the work should be clearly stated in the introduction as to how it differs from the cited work in reference 20. A brief discussion and references for impact of cyclodextrin inclusion complexes in governing chemical reactions for various real-world applications (https://doi.org/10.3389/fchem.2020.00641) should be included. The chemical structure of Indomethacin (IND) should be included in the introduction.

2) What is the aqueous solubility of indomethacin? Why was an acidic methanol (Meathanol:HCl) solution considered in the UV-vis studies of IND-βCD  complexes ?

3) In Figure 7, the physical mixtures with 1:1 and 1:0.5 seems to be missing the IND peak even at 0H of irradiation. Is there an explanation?

4) Have the photodegradation products of IND being largely characterized? If so, it’s better to list them along with the FTIR studies of the complex.

Author Response

Dear Professor,

I would like to express my thanks for Your assistance and opportunity to revise some parts of the manuscript entitled “Preparation and characterization of indomethacin supramolecular systems with β-cyclodextrin in order to estimate photostability improvement” by Marzena Jamrógiewicz and  Marek Józefowicz.

Please, refer to our modifications prepared in order to the comments. Language has also been approved. Performed corrections are assigned in yellow.

I sincerely thank for the time and effort in reviewing and correcting the manuscript and providing such important remarks, which definitely help to clarify the importance of this paper.

I hope that the revision will meet Yours and Editor’s expectations.

Yours sincerely,

Marzena Jamrógiewicz, PhD, D.Sc.

Explanations

  1. The novelty of the work should be clearly stated in the introduction as to how it differs from the cited work in reference 20. A brief discussion and references for impact of cyclodextrin inclusion complexes in governing chemical reactions for various real-world applications (https://doi.org/10.3389/fchem.2020.00641) should be included. The chemical structure of Indomethacin (IND) should be included in the introduction.

Thank you for the remark. We reedit the sentence in line 104: into: ” Research confirms that β-cyclodextrin has the most appropriate cavity size for IND, which may result in better stability and even increase solubility, what was approved by the studies in a liquid state.

The novelty and the aim of this work is to investigate the photostability of indomethacin, mainly in the form of supramolecular systems as inclusion complexes (IC) and physical mixtures (PM), in the solid phase and three different stoichiometries [1:1], [1:0.5] and [1:0.1].

  1. What is the aqueous solubility of indomethacin? Why was an acidic methanol (Meathanol:HCl) solution considered in the UVvis studies of IND-βCD complexes ?
  • Solubility of indomethacin in water is 0.0024 mg/mL and the value is pH dependent. We used recommended by Pharmacopoeia solvents mixture intended for indomethacin [European Pharmacopoeia 9.0, Indomethacin, 01/2017:0092, 2017].

  1. In Figure 7, the physical mixtures with 1:1 and 1:0.5 seems to be missing the IND peak even at 0H of irradiation. Is there an explanation?

In Figure 7 there are presented graphs of  heat capacity as a function of temperature for physical mixtures and inclusion complexes before and after photoirradiation. The graph obtained for pure IND is presented in Figure 6C. Physical mixtures with 1:1 and 1:0.5 (lower graph) present a significant “peak” near 157 °C.

  1. Have the photodegradation products of IND being largely characterized? If so, it’s better to list them along with the FTIR studies of the complex.
  • Thank you for that question. We are cautious that there are research in which photoproducts are identified. We are not able to identify them by the usage of FTIR only. For the determination of degradation products we would be forced to apply such tools as LC-MS/MS or GC-MS/MS as we previously did for different compound (ranitidine):

- Jamrógiewicz, M., & Wielgomas, B. (2013). Detection of some volatile degradation products released during photoexposition of ranitidine in a solid state. Journal of pharmaceutical and biomedical analysis, 76, 177-182.;

- Jamrógiewicz, M., Wielgomas, B., Strankowski, M. (2014). Evaluation of the photoprotective effect of β-cyclodextrin on the emission of volatile degradation products of ranitidine. Journal of pharmaceutical and biomedical analysis, 98, 113-119.

- Jamrógiewicz, M., Ciesielski, A. (2015). Application of vibrational spectroscopy, thermal analyses and X-ray diffraction in the rapid evaluation of the stability in solid-state of ranitidine, famotidine and cimetidine. Journal of pharmaceutical and biomedical analysis, 107, 236-243.

We are planning to prepare such a project.

For improve the presentation of FTIR application in the current research and indomethacin photostability improvement in supramolecular complexes we add the chemical groups into Figure 11 and 12, near narrows which places the changes of spectra.

Round 2

Reviewer 1 Report

I thank the authors for answering my questions and improving the manuscript. I also recommend that the authors pay attention in itnroduction to several recent review articles on supramolecular systems based on cyclodextrins for 2021. For example, Pharmaceutics 2021, 13(3), 409; Rus. Chem. Rev., 2021, 90(8), 895–1107; Biomolecules 2021, 11(3), 361